# Collagen/Chitosan Complexes: Preparation, Antioxidant Activity, Tyrosinase Inhibition Activity, and Melanin Synthesis

**DOI:** 10.3390/ijms21010313

**Published:** 2020-01-02

**Authors:** Yingying Hua, Chenjun Ma, Tiantian Wei, Liefeng Zhang, Jian Shen

**Affiliations:** 1School of Food Science and Pharmaceutical Engineering, Nanjing Normal University, Nanjing 210046, China; huaxueying1995@163.com (Y.H.); q443703814@126.com (C.M.); weitiantian1020@163.com (T.W.); 2Jiangsu Collaborative Innovation Center of Biomedical Functional Materials, Jiangsu Key Laboratory of Biomedical Materials, College of Chemistry and Materials Science, Nanjing Normal University, Nanjing 210046, China

**Keywords:** fish skin collagen, chitosan, antioxidant, inhibiting melanin synthesis

## Abstract

Bioactive collagen/chitosan complexes were prepared by an ion crosslinking method using fish skin collagen and chitosan solution as raw materials. Scanning electron microscopy observation confirmed that the collagen/chitosan complexes were of a uniform spherical shape and uniform particle size. The complexes were stable at different pH values for a certain period of time through swelling experiments. Differential scanning calorimetry (DSC) showed the collagen/ chitosan complexes were more stable than collagen. X-ray diffraction (XRD) showed that the complexes had a strong crystal structure, and Fourier transform infrared spectroscopy (FTIR) data revealed the changes in the secondary structure of the protein due to chitosan and TPP crosslinking. The content of malondialdehyde (MDA) in the complex treatment group was considerably lower, but the content of SOD was significantly higher than that of the collagen group or chitosan group. In addition, the collagen/chitosan complexes could considerably reduce melanin content, inhibit tyrosinase activity, and down-regulate tyrosinase mRNA expression. In conclusion, the collagen/chitosan complexes were potential oral protein preparation for antioxidant enhancement and inhibiting melanin synthesis.

## 1. Introduction

Collagen is the most abundant protein in animals. It is full of glycine, proline, and hydroxyproline [1]. At the same time, it has the characteristics of outstanding biocompatibility and degradation, low antigenicity [2,3,4,5,6]. The protein content of fish skin is higher than that of fish muscle, and the collagen content of fish skin can account for more than 80% of the total protein content, which is much higher than that of other parts of fish body [7]. The utilization of fish skin collagen not only solves the problem of fishery waste recovery, but also has the following advantages compared with mammalian collagen [6,8]. Firstly, fish skin collagen is easily soluble in neutral salt solution or dilute acid, even at low temperature, so it is easier to modulate soluble collagen solution [9,10]. Secondly, fish skin collagen has high bioactivity and safety, and is more likely to be absorbed by the human body [11]. Therefore, it shows a high application potential in functional food, pharmaceuticals, cosmetics, and other fields [12]. However, the thermal stability of fish skin collagen is relatively low, and it is vulnerable to thermal denaturation in the process of extraction and storage [13]. Therefore, how to improve the thermal stability and bioavailability of fish skin collagen has become a research hotspot.

Chitosan (CS) is a kind of positively charged chitin which is rich in nature. It has good biocompatibility, non-toxicity and biodegradability [14,15,16]. Chitosan is a kind of biopolymer widely used in biomedical field. It is often used in medical surgical dressings, tissue repair materials and drug-controlled release [17,18,19]. At the same time, when the pH value of chitosan is 6.5, it can also increase the acellular permeability of polypeptide drugs in mucosal epithelium [20]. This makes chitosan a more commonly used carrier for delivery of polypeptides and proteins.

Collagen/chitosan complexes have excellent physical and chemical properties and are widely used in biomedical fields. Collagen added to chitosan can obtain bionic structure similar to natural bone, increase the production of calcium and sulfate glycosaminoglycan, and have the potential of osteogenesis and cartilage formation [21,22]. The main application forms of collagen/chitosan complexes are membrane, gel, and sponge [23]. Materials based on chitosan and collagen can be used as bioactive substances that can affect cell behavior. Such materials promote the growth and differentiation of osteoblasts [24] and the proliferation of muscle cells [25]. The mechanical properties of the materials were improved by using the mixture of chitosan and collagen, demonstrating a higher resistance to enzymatic degradation [26,27,28]. Recent studies have also shown that the modification of collagen/chitosan complexes led to their high potential as drug delivery systems in the treatment of advanced cancer [29]. However, there is no relevant study on the antioxidant and melanin inhibitory activities of collagen/chitosan complexes.

The aim of the study is to evaluate the antioxidant and anti-melanin synthesis of collagen/chitosan complexes. The collagen/chitosan complex was prepared according to the special physical crosslinking method, then characterized by SEM, DSC, and FTIR. Furthermore, the antioxidant activity in vivo was explored. In addition, melanin content, tyrosinase activity, and tyrosinase mRNA expression were determined. This provides the experimental basis for the study of the antioxidant and melanin inhibition of collagen/chitosan complexes.

## 2. Results and Discussion

### 2.1. Preparation of the Collagen/Chitosan Complexes and Surface Morphology

The collagen/chitosan complex was prepared according to the sodium tripolyphosphate (TPP) ion crosslinking method. The chemical crosslinking of chitosan by crosslinkers such as glutaraldehyde, glyoxal, and glycol diglycidyl ether, is often not the first choice due to their physiological toxicity. The complex generation strategy was schematically illustrated in Scheme 1. Firstly, chitosan in acid solution is a cation, which can interact with the anion part of amphiprotic collagen molecule and form a complex through hydrogen bond. Chitosan is a polycation in acidic medium (pKa6.5), which can interact with the negatively charged TPP. It can be used to prepare cross-linked chitosan particles with good biocompatibility. Through electrostatic adsorption, the chitosan complex with collagen can react with TPP. Thus, with the aid of surfactant F68, a uniform and stable sphere was constituted.

SEM study was performed in order to directly observe the morphology of collagen/chitosan complexes, whereby the morphology of the complex was analyzed. The surface morphology of collagen, chitosan and collagen/chitosan complexes was shown in Figure 1. On the scale of 20 μm, collagen is composed of closely arranged, fine-grained, and interwoven collagen fibers, and the entire structure is lamellar. However, chitosan has a lamellar or spherical shell structure with small holes. The collagen/chitosan complexes formed by the interaction of the two materials had regular spherical shape and uniform particle size. This might be the stable spherical structure formed by the mixture of collagen and chitosan through the electrostatic interaction of TPP [30,31,32].

### 2.2. Swelling Properties

Figure 2 showed the swelling rate of the collagen/chitosan complexes immersed in phosphate buffered saline (pH = 1.2, 6.8, 7.4) within 3 h of the test. The results showed that the complex samples had high adsorption capacity for PBS with different pHs. The collagen/chitosan complex was incubated in pH 1.2 buffer for 0.5 h and reached the maximum swelling degree (33.58 ± 1.40%). The composite was more stable in pH 7.4 buffer and pH 6.8 buffer, which maximum swelling degree was 21.91 ± 3.99% and 24.32 ± 3.99% after incubation for 1 and 1.5 h. After this, the sample reached a slightly lower swelling ratio. The swelling degree of collagen/chitosan complexes was 14.94 ± 0.70% after incubation in 7.4 pH buffer for 3 h. The swelling degree of collagen/chitosan complexes was 16.72 ± 1.84% after incubation in pH 6.8 buffer for 3 h. The swelling degree of collagen/chitosan complexes was 10.39 ± 0.19% after incubation in 1.2 pH buffer for 3 h. The results showed that compared with the compounds in simulated intestinal fluid and body fluid, the compounds in simulated gastric acid condition reached the maximum swelling degree faster and had a lower swelling degree after 3 h. It may be that when the collagen/chitosan complexes were immersed in PBS buffer with strong acidity, water was more likely to penetrate into the complex and make it expand. With the hydration process, collagen bond broke and degraded. When the swelling reached the maximum value, the continuous degradation leaded to weight loss and the swelling degree decreases [33]. Meanwhile, the swelling ratio of the complexes was higher than 10% under PBS with three kinds of pH, and the first 1.5 h was stable. Therefore, the collagen/chitosan complex had better water absorption and swelling stability for PBS with different acid and alkali.

### 2.3. DSC and XRD Analysis

A differential scanning colorimeter (DSC) was used to heat chitosan, collagen, and collagen/chitosan complexes at the same time, so as to obtain a DSC thermogram and thermal transition temperature, and to determine whether the chemical and physical transition between chitosan and collagen occurred. As shown in Figure 3. Firstly, chitosan had an endothermic peak at 67.2 °C, which was produced by the decomposition of water in chitosan matrix [34]. At 27.22 °C, there was a peak of collagen thermal deformation temperature, and at 48.7 °C, there was a peak of collagen thermal contraction temperature [35]. When the collagen/chitosan complexes were at 27.22 °C, the small peak disappeared, which indicated that the collagen in the complex did not decompose. This might be due to the addition of chitosan to increase the thermal stability of collagen. At 62.6 °C, a new endothermic peak appeared in the complexes, indicating that the original endothermic peak of chitosan in the complex was influenced by collagen. In addition, it was found that there was an exothermic peak of collagen decomposition, but chitosan and collagen still did not reach the exothermic decomposition temperature at 102.6 °C. In conclusion, the collagen/ chitosan complexes prepared in this experiment were more stable than collagen.

In order to better understand the experimental collagen/chitosan complexes, X-ray diffraction analysis was carried out. Figure 4 showed the X-ray diffraction patterns of collagen, chitosan, and collagen/chitosan complexes of fish skin. There was a broad peak of collagen at 20.5° and no strong diffraction peak. The whole collagen showed amorphous structure [36]. Chitosan had two characteristic peaks at 13.56° and 19.96°. The reflection fell at 13.56° was assigned to crystal forms I. The strongest reflection appeared at 19.90°, which corresponded to crystal forms II [37]. However, in the complex, in addition to the characteristic peaks of chitosan at 19.10°, a new crystal diffraction peak appeared at 17.74°, 19.06°, 24.78°, 27.14°, and 30.90°. This showed that the structure changed during the formation of collagen/chitosan complexes, which made the whole complex exist in a crystal or molecular form.

### 2.4. Fourier Transform Infrared Spectrum (FTIR) Identification

FTIR spectra of collagen, chitosan and collagen/chitosan complexes were shown in Figure 5. Firstly, compared with collagen/chitosan complexes, the overlapping stretching vibration absorption peak of chitosan at –NH_2_ and –OH at 3431 cm^−1^ high frequency moved to 3310 cm^−1^. In addition, the amide I of chitosan adsorbed on the surface of the complex moved from 1655 to 1647 cm^−1^, while the amide II band moved from 1591 to 1535 cm^−1^. These results indicated that –NH_2_ and –OH groups in chitosan molecules participate in the reaction. Secondly, in the double bond region (2000–1500 cm^−1^), the C=O stretching vibration peak of fish skin collagen at 1583 cm^−1^ of amide I band was closely related to the secondary structure of protein. In the complex, the strong absorption peak of original collagen disappeared at 1583 cm^−1^. This indicated that the secondary structure of collagen might change when collagen was combined with chitosan. 

### 2.5. Antioxidation Activity In Vivo

Malonic dialdehyde (MDA) content is an important parameter reflecting the potential antioxidant capacity of an organism [38]. This can reflect the rate and intensity of lipid peroxidation and indirectly reflect the degree of tissue peroxidation damage [39]. Superoxide dismutase (SOD) has special physiological activity, and is the primary substance to scavenge free radicals in organisms [40]. The level of SOD in organism means the intuitive index of aging and death [41]. MDA content and SOD activity were collected in this study, and analysis of variance (ANOVA) was performed, as shown in Table 1. In this study, the aged mice were divided into the chitosan group, fish skin collagen group, collagen/chitosan complex group, and blank group, and a control group of young mice was designed. Twenty days later, MDA content and SOD activity in serum were detected by kit method. The results of variance analysis showed that the content of MDA increased significantly, and the activity of SOD decreased significantly in the aged mice groups compared with the young control group. This showed that with the increase of mice age, the degree of oxidation in mice also increased. The oxidation degree of old mice was significantly higher than that of young mice. Compared with the aged control group, the MDA content in collagen group and collagen/chitosan complexes group was very significantly lower (*p* < 0.01). The MDA content in chitosan group was significantly lower than that in the aged control group (*p* < 0.05). At the same time, SOD activity in the collagen group and collagen/chitosan complex group was very significantly higher than that in the aged control group (*p* < 0.01), and SOD activity in the chitosan group was significantly higher than that in the aged control group (*p* < 0.05). According to the significance analysis, the complexes group was better than the collagen group in reducing MDA content and increasing SOD activity. Chitosan, collagen, and complexes all had the effects of reducing MDA content and increasing SOD activity. Among them, the complexes had the best performance, which indicated that the complex prepared in this experiment had significant antioxidant effect and delaying aging in mice. The mechanism that the collagen/chitosan complexes had better antioxidant activity than collagen remains to be further studied.

### 2.6. Effect on Melanin Synthesis

#### 2.6.1. Determination of Melanin Content in B16 Melanoma Cells

Melanin is an important factor in determining the color of human skin, eyes and hair. Melanin synthesis is a stress response of melanocytes, which can cause melanosis, such as freckles, brown spots, melanoma, senile plaques and even melanocytoma [42,43]. Firstly, the relative content of melanin in B16 melanoma cells was detected. The effects of chitosan, fish skin collagen and collagen/chitosan complexes on melanin synthesis in B16 melanoma cells of mice were shown in Figure 6. Complex group and collagen group both inhibited melanin synthesis in B16 melanoma cells of mice, while chitosan group had little effect. When the mass concentration reached 12.5 µg/mL, the collagen/chitosan complexes group showed a significant inhibitory effect on melanin synthesis in B16 melanoma cells (*p* < 0.01), and its inhibitory ability increased with the increase of mass concentration, showing a significant dose-effect relationship. When the mass concentration reached 100 µg/mL, the relative content of melanin in the complex group was only (50.2 ± 4.6)% of that of the blank group, while that in the collagen group was (61.6 ± 5.4)% of that of the blank group. The inhibition effect of the complex group was better than that of the collagen group (*p* < 0.05). These results indicated that the ability of the collagen/chitosan complexes group to inhibit melanin synthesis was better than those of the collagen group and chitosan group.

#### 2.6.2. Detection of Tyrosinase Activity in B16 Melanoma Cells

Melanin biosynthesis constitutes a series of biochemical reactions initiated by tyrosinase-catalyzed tyrosine hydroxylation in vivo [44]. Tyrosinase plays a key role in a series of reactions of melanin biosynthesis. Therefore, controlling the activity of tyrosinase can control the amount of melanin produced. Secondly, the activity of tyrosinase in B16 melanoma cells was detected. The tyrosinase activity of B16 melanoma cells in mice was inhibited by chitosan, fish skin collagen and collagen/chitosan complexes at different concentrations as shown in Figure 7. At 12.5 µg/mL, collagen had no significant inhibitory effect on tyrosinase activity of melanoma cells (*p* > 0.05), but at 12.5 µg/mL, collagen/chitosan complexes had significant inhibitory effect on tyrosinase activity of melanoma cells (*p* < 0.05). At the same time, tyrosinase activity in the complex group was significantly inhibited at 50 and 100 µg/mL (*p* < 0.001). At 100 µg/mL, the tyrosinase activity in the complex group was only 61.8 ± 4.9% in the blank group, while that in the collagen group was 73.1 ± 4.7% in the blank group. The results showed that collagen/chitosan complexes could effectively inhibit tyrosinase activity in melanoma cells compared with collagen alone. This result may be related to the better biocompatibility of chitosan. The binding of collagen to chitosan not only increases stability, but also helps to absorb collagen.

#### 2.6.3. Detection of Tyrosinase mRNA Expression in B16 Melanoma Cells

Finally, tyrosinase mRNA expression in B16 melanoma cells was detected. According to the concentration data of tyrosinase activity influence experiment, the effects of collagen and collagen/chitosan complexes on tyrosinase mRNA expression in B16 melanoma cells were studied at 12.5, 25, 50, and 100 µg/mL concentrations. After 48 h, the results of reverse transcription polymerase chain reaction (RT-PCR) were shown in Figure 8. As can be seen from the figure, the DNA obtained by beta-actin RT-PCR was about 500 bp, and that obtained by tyrosinase RT-PCR was about 320 bp, which was consistent with the predicted results. All specimens had specific expression of beta-actin (500 bp band in Figure 8), which confirmed that the RNA was successfully extracted, and RT-PCR was normal. As shown in the figure 8, tyrosinase mRNA was expressed in murine B16 melanoma cells treated with different concentrations of collagen. With the increase of collagen concentration, tyrosinase gene expression in B16 melanoma cells decreased gradually. In the low concentration groups of 12.5 and 25 µg/mL, there was no significant difference in tyrosinase gene expression between B16 melanoma cells with the same concentration of the collagen/chitosan complexes and collagen. However, in 50 and 100 µg/mL medium and high concentration groups, the inhibition of tyrosinase by the complexes was stronger than that by collagen at the same concentration.

The results showed that different concentrations of collagen/chitosan complexes could significantly reduce melanin content, inhibit tyrosinase activity, and down-regulate tyrosinase mRNA expression. The collagen/chitosan complexes could inhibit tyrosinase expression more effectively than collagen. The inhibition of tyrosinase activity in B16 melanoma cells by collagen/chitosan complexes might be achieved by inhibiting the expression of its mRNA. From the mechanism of action, the collagen/chitosan complexes can effectively inhibit tyrosinase activity in melanoma cells, thus controlling the key step of melanin synthesis from tyrosine to dopaquinone and reducing the probability of melanin synthesis. However, the specific mechanism of this still needs to be further explored.

## 3. Materials and Methods

### 3.1. Materials

Fish skin collagen protein powder (MW < 3000 Da) was purchased from Dong KangYuan technology, Wuhan, China. Chitosan (deacetylation 0.75~0.85) and sodium tripolyphosphate (MW 367.86) were both provided by Jingchun Biochemistry Technology, Shanghai, China. Segmental polyether F-68 (MW 8350 ± 1000) was bought from Yuanye Biotechnology, Shanghai, China. All other solvents were of analytical or chromatographic grades and were commercially available.

### 3.2. Preparation of Chitosan Acetic Acid Solution

A certain amount of chitosan was dissolved in acetic acid solution with a concentration of 1% (v/v). Chitosan acetic acid solution with concentration of 2 mg/mL was obtained by continuous stirring at room temperature and medium speed for 1 h. Then chitosan acetic acid solution was obtained by placing it on a magnetic stirrer and swelling overnight at room temperature. The chitosan acetic acid solution was adjusted to pH 5 for use.

### 3.3. Complex Preparation

The collagen/chitosan complex was prepared by ion crosslinking of sodium tripolyphosphate (TPP) [45]. A proper amount of fish skin collagen was dissolved in 0.1 M NaOH solution to prepare 10 mg/mL solution. Take 2 mL collagen solution drop by drop and add 4mL chitosan acetic acid solution to stir. The solution was stirred at room temperature with a magnetic stirrer at medium speed for 2 h. After that, 0.5 mL surfactant Pluronic F68 solution (1 mg/mL) was added to it. When feeble emulsification occurs in the system, proper amount of sodium tripolyphosphate (1 mg/mL) is slowly added under stirring and stirred continuously for 2 h at medium speed. Then, under the centrifugal condition of 13,500 rpm, 4 °C, the solution was centrifugated 1 h to obtain the precipitation of the complex. Finally, the collagen/chitosan complexes precipitation was cleaned twice with deionized water and freeze-dried for 24 h to obtain the complex drying substance, which was stored in 4 °C for reserve.

### 3.4. Physicochemical Characterization of Complex

#### 3.4.1. Scanning Electron Microscopy

Scanning electron microscopy (SEM, JSM-6510, Japan Electronics Corporation, Tokyo, Japan) was used to observe the morphology of chitosan fish skin collagen and the collagen/chitosan complexes. The conductive adhesive was bonded to the sample base, and the collagen and collagen/chitosan complexes powder was evenly sprayed on the conductive adhesive. The unbonded powders were blown away with ear wash balls, and gold was sputtered for 2 min with ion plating apparatus.

#### 3.4.2. Swelling Properties

The 10mg of dried collagen/chitosan complexes sample (*n* = 3) were dissolved in 1mL PBS buffer (pH = 1.2, 6.8, 7.4) and placed in the shaking table of 37 °C and 135 r/min. The supernatant was removed by centrifugation (8000 r/min, 5 min) every 30min. After the surface moisture is absorbed, the weight of the complex after water absorption is weighed and observed for 3 h continuously. The water absorption expansion rate is calculated according to the following formula:
Water absorption expansion rate% = ((W_w_ − W_d_)/W_d_) × 100%
where W_w_ refers to the weight of the collagen/chitosan complexes after water absorption and W_d_: the weight of dried collagen/chitosan complexes sample.

#### 3.4.3. Differential Scanning Calorimetry

Complex powder, chitosan powder, and fish skin collagen powder were taken in a proper amount and scanned in the range of 25 to 150 °C by differential scanning calorimetry (DSC, Shimadzu DSC-50 system, Shimadzu, Kyoto, Japan) under a nitrogen environment at a heating rate of 10 °C/min. A thermal analysis pattern was drawn.

#### 3.4.4. X-ray Diffraction Analysis

The collagen/chitosan complexes powder, chitosan powder and fish skin collagen powder were scanned by X-ray diffractometer (tube voltage 40 KV, Cu-Ka radiation, current 200 mA, scanning speed 2 degree/min, scanning range 3–45 degree, Rigaku Corporation, Tokyo, Japan), and the X-ray diffraction pattern was drawn.

#### 3.4.5. Fourier Transform Infrared Spectrum Identification

The KBr tablets were prepared by mixing 5 mg of the collagen/chitosan complexes powder, chitosan powder and fish skin collagen powder with appropriate amount of KBr. The scanning range was 400–4000 cm^−1^ and the resolution was 2 cm^−1^. The infrared spectra of the collagen/chitosan complexes, chitosan, and fish skin collagen were drawn (Nexus 670, Nicolet, Woodland, CA, USA).

### 3.5. Antioxidant Activity In Vivo

Ten-month old mice (40 numbers), clean grade, weighing 40.0 ± 5.0 g, half male and half female; three-month old mice (10 numbers), clean grade, weighing 25.0 ± 3.4 g, were purchased from Qinglongshan Animal Breeding Farm, Jiangning District, Nanjing, China. The feeding conditions were 22 ± 2 °C, 50 ± 5% relative humidity, 12 h of light/12 h of darkness. The mice were fed standard food, free drinking water and adapted to the environment for 7 days before the experiment.

Male and female old mice were randomly divided into four groups (10 mice/group): blank group (Ultra pure water 25 mL/(kg·day)), chitosan (200 mg/(kg·day)), fish skin collagen (200 mg/(kg·day)) and collagen/chitosan group (400 mg/(kg·day)). Another half male and half female young mice (10 numbers), were selected as the control group (Ultra-pure water 25 mL/(kg·day)). All the samples were administered orally by gavage (No.12 gastric perfusion needle and 1.0 mL syringe, kindly, Shanghai, China). All groups were administered once a day for 20 consecutive days according to the above methods. Two hours after the last experimental operation, orbital blood was taken from mice and serum was separated by centrifugation. The content of malondialdehyde in serum was determined by TBA colorimetry (Malondialdehyde assay kit, Jiancheng Bioengineering Institute, Nanjing, China) and the activity of SOD in serum was determined by xanthine oxidase (Superoxide Dismutase assay kit, Jiancheng Bioengineering Institute, Nanjing, China). All the experimental treatments followed the National Institute of Health’s Guide for the Care and Use of Laboratory Animals.

### 3.6. Effect of Melanin Synthesis

#### 3.6.1. Determination of Melanin Content

B16 melanoma cells were cultured in logarithmic growth phase in RPMI-1640 medium (Gibco, New York, NY, USA) and diluted to 1 × 10^5^/mL. The cells (400 µL) were inoculated into 24-well plates per well for 24 h at 37 °C. Then, the culture medium was discarded, and the concentrations of 12.5, 25, 50, and 100 µg/mL (prepared with RPMI-1640) collagen and collagen/chitosan complexes solution were added in the amount of 400 µL per pore, while the blank group was set up with 400 µL serum-free pure medium instead of the determination solution containing collagen or collagen/chitosan complexes. Each concentration gradient group was cultured for 48 h in triplicate. The cells were digested and centrifuged. The supernatant was discarded and washed with PBS twice. Then, the precipitation was blown and dispersed by 120 μL 1 M NaOH (containing 10% DMSO). Then it was incubated at 90 °C for 30 min. 100 μL lysate was absorbed into 96-well plate. The relative melanin content was determined by ELISA reader (Bio-Rad, Berkeley, CA, USA) at 405 nm [46,47].
the relative content of melanin in cells% = (As/Ac) × 100%
As refers to the absorptivity of collagen, chitosan or the collagen/chitosan complexes, and Ac refers to light absorption of blank test group.

#### 3.6.2. Detection of Tyrosinase Activity

Tyrosinase can rapidly oxidize L-tyrosine to L-3,4-dihydroxyphenylalanine (L-DOPA) and further convert it to dopaquinone, which is the first stage of melanin biosynthesis. The two-step reaction is catalyzed by tyrosinase, and the reaction is rapid. The absorbance of dopaquinone at 475 nm is linearly related to its concentration. Therefore, the activity of tyrosinase determines the amount of brown dopaquinone [48]. B16 melanoma cells were cultured in logarithmic growth phase in RPMI-1640 medium (Gibco, New York, NY, USA) and diluted to 1 × 10^5^/mL. The cells (400 µL) were inoculated into 96-well plates per well for 24 h at 37 °C. Then, the culture medium was discarded, and the concentration of 0, 12.5, 25, 50 and 100 µg/mL (prepared with RPMI-1640) collagen and collagen/chitosan complex solution were added in the amount of 400 µL per pore, and the blank group was set up with 400 µL serum-free pure medium instead of the determination solution containing collagen or collagen/chitosan complexes. Each concentration gradient group were collected 48 h after culture in triplicate. Then, the cells were washed twice with 0.1 mol/L phosphate buffer (pH 6.8). Triton X/PBS buffer solution (1%, volume fraction) was added to each hole, and placed in refrigerator at −80 °C for 30 min. After melting, 10 µL L-DOPA (10 mM) solution was added at 37 °C and kept for 30 min. The absorbance of the reaction solution at 475 nm was measured by a spectrophotometer (Bio-Rad, Berkeley, CA, USA), and the relative activity of tyrosinase in melanoma cells was calculated [49,50].
Relative activity of tyrosinase% = [(A_30_ − A_0_)/(A_C30_ − A_C0_)] × 100%
A_0_ and A_30_ refers to the absorbency values of collagen, chitosan or collagen/chitosan complexes test solution in 0 and 30 min. A_C0_ and A_C30_ refers to the absorbency values of the blank group in 0 and 30 min.

#### 3.6.3. Detection of Tyrosinase mRNA Expression

Total RNA extraction: Trizol reagent was used to extract total RNA according to the instructions of the Trizol kit (Invitrogen, Waltham, MA, USA). B16 melanoma cells were cultured in logarithmic growth phase in RPMI-1640 medium (Gibco, New York, NY, USA) and diluted to 1 × 10^5^/mL. The cells (400 µL) were inoculated into 24-well plates per well for 24 h at 37 °C. Then, the culture medium was discarded, and the concentration of 12.5, 25, 50 and 100 µg/mL (prepared with RPMI-1640) collagen and collagen/chitosan complexes solution were added in the amount of 400 µL per pore, and the blank group was set up with 400 µL serum-free pure medium instead of the determination solution containing collagen or collagen/chitosan complexes. Each concentration gradient group was cultured for 48 h in triplicate. Suck up the culture medium, add 0.25 mL Trizol to each hole of 24-hole plate. Shake for 3–5 times, blow for 2–3 times with gun to ensure all cracking, and then suck into the centrifugal pipe. Leave it at room temperature for 5 min, so that the sample can be fully cracked. Add 0.2 mL of chloroform to 1 mL of Trizol, shake vigorously for 15 s, and place at room temperature for 2–3 min. Centrifugation at 4 °C at 12,000 rpm for 15 min, then pipette the upper colorless water phase containing total RNA into a new centrifuge tube, add 0.5 mL isopropanol, mix it upside down several times, and precipitate at room temperature for 10 min. After centrifugation at 12,000 rpm for 10 min, RNA precipitates were found at the bottom of the tube and the supernatant was discarded. Add 1 mL of 75% ethanol (prepared with DEPC water), mix it upside down, centrifugate at 7500 rpm 4 °C for 5 min, and discard the supernatant. Shake it with a centrifuge (>5000 rpm, centrifugation for 1 s) and carefully suck up the liquid. After drying the volatile ethanol for 10 min, add 20 µL DEPC water to dissolve it, and place it at room temperature for 10min. Put it on ice and measure the RNA concentration.

RT-PCR: Primer design (Shanghai Institute of Cells, Chinese Academy of Sciences): Mouse tyrosinase: (+) 5′-TTC AAA GGGG TG GATGAC CG-3′, (−) 5′-GAC ACA TAG TAA TGC ATC-3′, the expected length of amplified fragment is 319 bp; Mouse beta-actin: (+) 5′-TCAGAA GGA CTC CTA TGG-3′, (−) 5′-TCT TTA TCG CA CG-3′, the expected length of amplified fragment is 500 bp. Take 1 µL mRNA, 4 µL of one step RT-PCR mix premix (Biotechrabbit, Shanghai, China), and add water to make the total volume of reaction mixture 25 µL. The total volume of the reaction mixture was 25 µL. The optimum conditions for PCR were 94 °C 1 min, 94 °C 1 min, 54 °C 1 min, 72 °C 1 min, 33 cycles, and 72 °C extension for 6 min.

PCR product analysis: 10 µL PCR amplified products and 1000 bp Marker (TaKaRa Bio Group, Dalian, China) was added to the 2% agarose gel containing 0.5 mL ethidium bromide, and electrophoresis was carried out for 30 min. The IS-3400 gel imaging analysis system (Alpha, Beijing, China) was applied to gel imaging analysis to observe the results.

### 3.7. Statistical Analysis

All experiments were repeated at least three times. Results were expressed as mean values and standard error (mean ± SE). Data were compared using a one-way analysis of variance (ANOVA) to evaluate the statistical significance. Post-hoc Tukey’s test was used to compare the significance of deviations in the measured data of each group. In all cases, differences were considered statistically significant at *p* < 0.05.

## 4. Conclusions

In this study, the collagen/chitosan complexes were successfully prepared, which had steady sheet structure and good antioxidant activity in vivo. Melanin content, tyrosinase activity, and tyrosinase mRNA expression in B16 melanoma cells of mice administered were determined. The results showed that compared with collagen, collagen/chitosan complexes could significantly inhibit melanin synthesis, reduce tyrosinase activity and down-regulate tyrosinase mRNA expression. In this study, the basic physical and chemical properties and biological activities of collagen/chitosan complexes were discussed. This provided preliminary reference for further research on the application of collagen/chitosan complexes in the field of skin protection in the future. However, the mechanism of improving the bioactivity of collagen/chitosan complex remains to be further elucidated.

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
