# Peer review of "Collagen/Chitosan Complexes: Preparation, Antioxidant Activity, Tyrosinase Inhibition Activity, and Melanin Synthesis"

_ijms, 2020, doi:10.3390/ijms21010313_

Round 1
Reviewer 1 Report
The manuscript underwent an extensive editing, with several additions and amendements. The content is now satisfactory, with no major shortcomings, but the English language remain rather shaky, beginning with the new title.
Author Response
Review1:
The manuscript underwent an extensive editing, with several additions and amendements. The content is now satisfactory, with no major shortcomings, but the English language remain rather shaky, beginning with the new title.
Answer: Thank you for your suggestion. We have modified all languages of the manuscript including the title and highlighted it in red.
Reviewer 2 Report
The authors of the publication explained all the comments of the reviewer. The mechanism of enhancing the bioactivity of collagen / chitosan complex has been further elaborated in the revised manuscript. This issue, initially discussed in the publication, requires further research. The mechanism proposal is sufficient at this stage. The authors introduced changes to the text, which led to the fact that in this form I recommend publication for publication.
Author Response
Review2:
The authors of the publication explained all the comments of the reviewer. The mechanism of enhancing the bioactivity of collagen / chitosan complex has been further elaborated in the revised manuscript. This issue, initially discussed in the publication, requires further research. The mechanism proposal is sufficient at this stage. The authors introduced changes to the text, which led to the fact that in this form I recommend publication for publication.
Answer: Thank you for your confirmation and suggestions.
Reviewer 3 Report
The manuscript is now well improved; however, some major issues are still addressed.
General Comments:
Please refine the language and check the grammar mistake very carefully throughout MS.
Still, some experiments are incomplete, should provide the data of three samples in all experiments and should explain the proper reason for a particular study design especially in animal experiments.
Specific Comments:
Line 17. (DSC), X-ray diffraction (XRD)..(FTIR) were used to identify stable chemical structure: Readers already know the application of these techniques so no need to explain, the author should write the results/observation got from these experiments. For instance: FTIR data revealed the changes in the secondary structure of protein due to chitosan and TPP crosslinking.
Line 60 delete “further”
Line 70 Chemical crosslinking of chitosan, such as: Chemical crosslinking of chitosan by crosslinkers such as..
Line 72.It can be used to prepare cross- linked chitosan particles with good biocompatibility.: The authors discuss only TPP and chitosan, should discuss the interaction between chitosan and collagen; collagen and TPP; and how TPP crosslinks both chitosan and collagen.
Line 80 closely arranged, thin in diameter and interwoven into a network with typical characteristics of type I collagen and triple helix structure,: This observation is not enough or inappropriate to conclude the features like type I and triple helix structure.
with typical characteristics of type I collagen and triple helix structure: Where is it seen in Fig.1a?
chitosan presented a small pore structure: What is the flat like structure with holes behind the globules. Is it also chitosan?
Line 83 mght be the: might be the
Line 92-96. The maximum swelling degree of … after incubation for 0.5 h in pH 1.2 buffer: Combine in one sentence.
Line 111. Figure 2.:Collagen (alone) and chitosan (alone) results missing in this experiment
Line 117. Mention the figure number. Firstly, chitosan had an endothermic peak at 67.2 oC (Fig. 3),….
Looks like the green box noted place is the maximum peak, check (Refer the attachment for details).
Line 168. aged mice groups compared with the young control group: Explain the main reason for choosing young and old age mice in this experiment.
Line 183 Table 1. contain same amount of collagen and chitosan : This means 200g/kg.bw of collagen and 200g/kg.bw of chitosan? So 400g/kg.bw of total complex sample administered? Give the specific dosage of t complex sample.
Line 244 Figure 8. Expression of tyrosinase gene in B16 melanoma cells. (M) Marker, (A) Control, (B-E) Collagen 12.5, 25, 50, 100 μg/ml; (F-I) Collagen/chitosan complexes 12.5, 25, 50, 100 μg/ml.: Data for Chitosan Group were missing here. The maker band is hard to observe. Do the amendment.
304 a little of the collagen/chitosan complexes powder: 10 mg of the collagen/chitosan complexes powder…Write the exact sample quantity like this.
Line 309. 40 aged (10 months old) mice: Change to “Ten months old mice (40 numbers)”…
Lines 314-315 Forty aged mice were randomly divided into four groups. There were 10 mice in each group, half male and half female. They were blank aging control group: Change to “Male and female mice were randomly divided into four groups (10 mice/group):control (give details, saline?), chitosan (200 mg/kg.d), fish skin collagen (200 mg/kg.d) and collagen/chitosan group (200 mg/kg.d). All the samples were administered orally by gavage (give the details of gavage size, brand, company)”. The author should refine this manuscript like the above.
Lines 317-320. The mice in the chitosan group were given 200 mg/(kg·d) …content of collagen by gavage.: Delete these sentences
Another 10 young mice, half male and…: Confusing, the authors mentioned 40 mice and divided into four groups (10 per group), then how they selected another 10 mice? So a total of 50 mice? What is the blank aging control group and control group? Are they same or different?
Line 366 Total RNA extraction: Trizol reagent: First explain how the cells were treated with your samples for mRNA experiment then discuss RNA extraction protocol.
The author should provide the microscopic (wither light or fluorescent images) images of B16 melanoma cells treated with and without your samples. Because there is no evidence that the authors used these cells in their experiments.
If possible provide some pictures of 1) animal experiments (some animal photos during the experiment) and 2) collagen/chitosan complexes to help the reader to understand your work more clearly.
The conclusion part is now convincing.
Next time, Kindly Provide a separate attachment, describing the authors’ response to all the reviewers’ comments for easy tracking. It is very hard to cross-check the revision.

Author Response
Review3:
General Comments:
Please refine the language and check the grammar mistake very carefully throughout MS.
Still, some experiments are incomplete, should provide the data of three samples in all experiments and should explain the proper reason for a particular study design especially in animal experiments.
Answer: Thank you for your suggestion. We have optimized the language expression of the manuscript, seriously revised the improper grammar in the manuscript, and marked it with red marks.
At present, there are two experiments without three sample groups.
Swelling properties.
We have studied the relative stability of collagen / chitosan complex, and the swelling properties of chitosan and collagen at different pH values have been tested in previous articles, and we have no further experiments to prove it. Specific references are as follows:
Ren, Dongwen , et al. "The enzymatic degradation and swelling properties of chitosan matrices with different degrees of N-acetylation." 340.15(2005):2403-2410.
Kozlowska, Justyna , and A. Kaczmarkiewicz . "Collagen matrices containing poly(vinyl alcohol) microcapsules with retinyl palmitate – Structure, stability, mechanical and swelling properties." Polymer Degradation and Stability 161(2019):108-113.
Yang, Chunrong . "Enhanced physicochemical properties of collagen by using EDC/NHS-crosslinking." Bulletin of Materials Science 35.5(2012):913-918.
Detection of tyrosinase mRNA expression in B16 melanoma cells
In the design of the experiment, it was considered that chitosan had little effect on tyrosine kinase activity at low concentration. Specific references are as follows. The results in Figure 7 also showed that chitosan had no effect on tyrosine kinase activity in this concentration range. Therefore, the chitosan group was not considered in the experimental design. If we want to design the experiment in the future, we must consider it comprehensively.
Wen Chen, et al. "Inhibition of melanin synthesis and melanosome transfer by chitosan biomaterials." Journal of Biomedical Materials Research Part B Applied Biomaterials 4(2019):1-12.
In addition, in the animal experiment, a positive control and a negative control were added in addition to the three sample groups.
Specific Comments:
Line 17. (DSC), X-ray diffraction (XRD)..(FTIR) were used to identify stable chemical structure: Readers already know the application of these techniques so no need to explain, the author should write the results/observation got from these experiments. For instance: FTIR data revealed the changes in the secondary structure of protein due to chitosan and TPP crosslinking.
Answer: Thank you for your suggestion. We have modified the summary part of the manual change to change
"Differential scanning calorimetry (DSC), X-ray diffraction (XRD) and Fourier transform infrared spectroscopy (FTIR) were used to establish stable chemical structure formed between chitosan and fish skin collagen. Furthermore, the collagen/chitosan complexes had excellent antioxidant activity." to " Differential scanning calorimetry (DSC) showed the collagen/ chitosan complexes were more stable than collagen. The X-ray diffraction (XRD) showed that the complexes had a strong crystal structure, and the Fourier transform infrared spectroscopy (FTIR) data revealed the changes in the secondary structure of protein due to chitosan and TPP crosslinking."
Line 60 delete “further”
Answer: Thank you for your suggestion. We have deleted " further" in the revised manuscript.
Line 70 Chemical crosslinking of chitosan, such as: Chemical crosslinking of chitosan by crosslinkers such as..
Answer: Thank you for your suggestion. We have changed "Chemical crosslinking of chitosan by crosslinkers such as" to "Chemical crosslinking of chitosan by crosslinkers such as" in the manuscript.
Line 72.It can be used to prepare cross- linked chitosan particles with good biocompatibility.: The authors discuss only TPP and chitosan, should discuss the interaction between chitosan and collagen; collagen and TPP; and how TPP crosslinks both chitosan and collagen.
Answer: Thank you for your suggestion. We have supplemented this paragraph in the manuscript with the following details:
" The collagen/chitosan complex was prepared by sodium tripolyphosphate (TPP) ion crosslinking method. Chemical crosslinking of chitosan by crosslinkers such as glutaraldehyde, glyoxal and glycol diglycidyl ether, is often not the first choice because of their physiological toxicity. The complex generation strategy was schematically illustrated in Scheme 1. Firstly, chitosan in acid solution is a cation, which can interact with the anion part of amphiprotic collagen molecule and form a complex through hydrogen bond. Chitosan is a polycation in acidic medium (pKa6.5), which can interact with negatively charged TPP. It can be used to prepare cross-linked chitosan particles with good biocompatibility. Through electrostatic adsorption, the chitosan complex with collagen can react with TPP. Then, with the aid of surfactant F68, a uniform and stable sphere was constituted."
Line 80 closely arranged, thin in diameter and interwoven into a network with typical characteristics of type I collagen and triple helix structure,: This observation is not enough or inappropriate to conclude the features like type I and triple helix structure.
with typical characteristics of type I collagen and triple helix structure: Where is it seen in Fig.1a?
chitosan presented a small pore structure: What is the flat like structure with holes behind the globules. Is it also chitosan?
Answer: Thank you for your correction. The inappropriate expression in the article has been modified.The flat structure shown in the picture of chitosan is also a part of chitosan.The manuscript was modified as follows: "On the scale of 20 μ m, collagen is composed of closely arranged, fine-grained and interwoven collagen fibers, and the whole structure is lamellar. However, chitosan has a lamellar or spherical shell structure with small holes."
Line 83 mght be the: might be the
Answer: Thank you for your correction. We are very sorry for our mistake. We have revised it in the article.
Line 92-96. The maximum swelling degree of … after incubation for 0.5 h in pH 1.2 buffer: Combine in one sentence.
Answer: Thank you for your suggestion. We have revised this sentence in the manuscript as follows: "The collagen/chitosan complex was incubated in pH 1.2 buffer for 0.5 h and reached the maximum swelling degree (33.58±1.40%). The composite was more stable in pH 7.4 buffer and pH 6.8 buffer, which maximum swelling degree was 21.91±3.99% and 24.32±3.99% after incubation for 1 and 1.5 h."
Line 111. Figure 2.:Collagen (alone) and chitosan (alone) results missing in this experiment
Answer: Thank you for your suggestion. We have studied the relative stability of collagen / chitosan complex, and the swelling properties of chitosan and collagen at different pH values have been tested in previous articles, and we have no further experiments to prove it. Specific references are as follows:
Ren, Dongwen , et al. "The enzymatic degradation and swelling properties of chitosan matrices with different degrees of N-acetylation." 340.15(2005):2403-2410.
Kozlowska, Justyna , and A. Kaczmarkiewicz . "Collagen matrices containing poly(vinyl alcohol) microcapsules with retinyl palmitate – Structure, stability, mechanical and swelling properties." Polymer Degradation and Stability 161(2019):108-113.
Yang, Chunrong . "Enhanced physicochemical properties of collagen by using EDC/NHS-crosslinking." Bulletin of Materials Science 35.5(2012):913-918.
Line 117. Mention the figure number. Firstly, chitosan had an endothermic peak at 67.2 oC (Fig. 3),….
Looks like the green box noted place is the maximum peak, check (Refer the attachment for details).
Answer: Thank you for your suggestion. The maximum peak in the green box indicated in your attachment is the maximum peak of the complex, not the maximum peak of chitosan. The maximum value of chitosan can be seen clearly from the figure that it is shifted to the direction of high temperature. Through the analysis of origin86 software, the maximum absorption peak of chitosan was 67.2oC.
Line 168. aged mice groups compared with the young control group: Explain the main reason for choosing young and old age mice in this experiment.
Answer: Thank you for your suggestion. We use young mice here to prove that they have high SOD value and low MDA value when they are young. The comparison between the old group and the young group is to show that the oxidation-related indicators will increase with the increase of age. Therefore, the drug administration experiment for the elderly group is scientific and accurate. We have made changes in the manuscript as follows: "The results of variance analysis showed that the content of MDA increased significantly and the activity of SOD decreased significantly in the aged mice groups compared with the young control group. This shows that with the increase of mice age, the degree of oxidation in mice is also increasing. The oxidation degree of old mice was significantly higher than that of young mice."
Line 183 Table 1. contain same amount of collagen and chitosan : This means 200g/kg.bw of collagen and 200g/kg.bw of chitosan? So 400g/kg.bw of total complex sample administered? Give the specific dosage of t complex sample.
Answer: Thank you for your suggestion. We have added the specific amount of drugs to the manuscript, and the specific changes are as follows.
Group |
Dose (mg/(kg·d)) |
Animal n |
MDA nmol/ml |
SOD ×103Nu/ml |
Young control |
- |
10 |
5.68 ± 1.23** |
7.64 ± 0.87** |
Aged control |
- |
10 |
10.24 ± 1.15 |
5.57 ± 0.38 |
Chitosan |
200 |
10 |
8.81 ± 1.06* |
6.17 ± 0.12* |
Collagen |
200 |
10 |
8.03 ± 1.46** |
6.54 ± 0.57** |
Complex |
400 (contain same amount of collagen and chitosan) |
10 |
7.61 ± 1.43** |
6.77 ± 0.33** |
Line 244 Figure 8. Expression of tyrosinase gene in B16 melanoma cells. (M) Marker, (A) Control, (B-E) Collagen 12.5, 25, 50, 100 μg/ml; (F-I) Collagen/chitosan complexes 12.5, 25, 50, 100 μg/ml.: Data for Chitosan Group were missing here. The maker band is hard to observe. Do the amendment.
Answer: Thank you for your correction. In the design of the experiment, it was considered that chitosan had little effect on tyrosine kinase activity at low concentration. Specific references are as follows. The results in Figure 7 also showed that chitosan had no effect on tyrosine kinase activity in this concentration range. Therefore, the chitosan group was not considered in the experimental design. If we want to design the experiment in the future, we must consider it comprehensively.
Wen Chen, et al. "Inhibition of melanin synthesis and melanosome transfer by chitosan biomaterials." Journal of Biomedical Materials Research Part B Applied Biomaterials 4(2019):1-12.
304 a little of the collagen/chitosan complexes powder: 10 mg of the collagen/chitosan complexes powder…Write the exact sample quantity like this.
Answer: Thank you for your suggestion. We have revised it in the manuscript. The specific changes are as follows: "5 mg of the collagen/chitosan complexes powder "
Line 309. 40 aged (10 months old) mice: Change to “Ten months old mice (40 numbers)”…
Answer: Thank you for your suggestion. We have revised it in the manuscript. The specific changes are as follows: "Ten months old mice (40 numbers), clean grade, weighing 40.0 ± 5.0 g, half male and half female; three months old mice (10 numbers), clean grade, weighing 25.0 ± 3 g "
Lines 314-315 Forty aged mice were randomly divided into four groups. There were 10 mice in each group, half male and half female. They were blank aging control group: Change to “Male and female mice were randomly divided into four groups (10 mice/group):control (give details, saline?), chitosan (200 mg/kg.d), fish skin collagen (200 mg/kg.d) and collagen/chitosan group (200 mg/kg.d). All the samples were administered orally by gavage (give the details of gavage size, brand, company)”. The author should refine this manuscript like the above.
Answer: Thank you for your suggestion. We have revised it in the manuscript. The specific changes are as follows: "Male and female old mice were randomly divided into four groups (10 mice/group): blank group (Ultra pure water 25 mL/(kg·d)), chitosan (200 mg/(kg·d)), fish skin collagen (200 mg/(kg·d)) and collagen/chitosan group (400 mg/(kg·d)). Another half male and half female young mice (10 numbers), were selected as the control group (Ultra pure water 25 mL/(kg·d)). All the samples were administered orally by gavage (No.12 gastric perfusion needle and 1.0 ml syringe, kindly, Shanghai)."
Lines 317-320. The mice in the chitosan group were given 200 mg/(kg•d) …content of collagen by gavage.: Delete these sentences
Another 10 young mice, half male and…: Confusing, the authors mentioned 40 mice and divided into four groups (10 per group), then how they selected another 10 mice? So a total of 50 mice? What is the blank aging control group and control group? Are they same or different?
Answer: Thank you for your advice. We have made corresponding modifications to the manuscript, and the relevant statements have been deleted. In this paper, there are 50 experimental mice, 10 young mice and 40 old mice. The young mice were used as the control group, and the old mice in the blank group were given ultrapure water (25 mL/(kg·d)). Except for the age difference, the other factors were the same in the young mice and the old blank mice.
Line 366 Total RNA extraction: Trizol reagent: First explain how the cells were treated with your samples for mRNA experiment then discuss RNA extraction protocol.
Answer: Thank you for your advice. We have supplemented the experimental steps with the details as follows: "B16 melanoma cells were cultured in logarithmic growth phase in RPMI-1640 medium (Gibco, New York, USA) and diluted to 1×105/mL. The cells (400 µL) were inoculated into 24-well plates per well for 24 hours at 37oC. Then the culture medium was discarded, and the concentration of 12.5, 25, 50 and 100 µg/mL (prepared with RPMI-1640) collagen and collagen/chitosan complexes solution were added in the amount of 400 µL per pore, and the blank group was set up with 400 µL serum-free pure medium instead of the determination solution containing collagen or collagen/chitosan complexes. Each concentration gradient group was cultured for 48 hours in triplicate. Suck up the culture medium, add 0.25 mL Trizol to each hole of 24 hole plate. Shake for 3-5 times, blow for 2-3 times with gun to ensure all cracking, and then suck into the centrifugal pipe. Leave it at room temperature for 5 minutes, so that the sample can be fully cracked. Add 0.2 mL of chloroform to 1mL of Trizol, shake vigorously for 15 seconds, and place at room temperature for 2-3 minutes. Centrifugation at 4oC at 12000 rpm for 15 minutes, then pipette the upper colorless water phase containing total RNA into a new centrifuge tube, add 0.5 mL isopropanol, mix it upside down several times, and precipitate at room temperature for 10 minutes. After centrifugation at 12000 rpm for 10 minutes, RNA precipitates were found at the bottom of the tube and the supernatant was discarded. Add 1 mL of 75% ethanol (prepared with DEPC water), mix it upside down, centrifugate at 7500 rpm 4oC for 5 minutes, and discard the supernatant. Shake it with a centrifuge (> 5000 rpm, centrifugation for 1s) and carefully suck up the liquid. After drying the volatile ethanol for 5-10 min, add 20 µL DEPC water to dissolve it, and place it at room temperature for 10min. Put it on ice and measure the RNA concentration."
The author should provide the microscopic (wither light or fluorescent images) images of B16 melanoma cells treated with and without your samples. Because there is no evidence that the authors used these cells in their experiments.
Answer: Thank you for your suggestion. I'm sorry, we didn't design the cell level fluorescence microscope experiment, so we can't give the relevant cell image. If there is a chance to design the relevant fluorescence microscopy experiment in the future, it will make the experiment more convincing. Here, some optical micrographs of B16 cells used in this experiment can be provided. The specific images are as follows:
If possible provide some pictures of 1) animal experiments (some animal photos during the experiment) and 2) collagen/chitosan complexes to help the reader to understand your work more clearly.
Answer: Thank you for your suggestion. The specific images are as follows:
|
|
|
Figure (1-1) and figure (1-2) are the animal experiments’ pictures; figure (2-1) is the collagen/chitosan complexes’ picture.
The conclusion part is now convincing.
Next time, Kindly Provide a separate attachment, describing the authors’ response to all the reviewers’ comments for easy tracking. It is very hard to cross-check the revision.
Answer: Thank you for your advice. All changes have been reflected separately in response to reviewer 3.

Round 2
Reviewer 3 Report
Dear Authors
Thank you so much for the excellent revision and great efforts. I hope my comments are useful to improve the MS standard. Your response is very convincing. I am very pleased to accept your manuscript for publication. Great work!